

# Secure biometric authentication with de-duplication on distributed cloud storage

Vinoth Kumar M[1], K Venkatachalam[2], Prabu P[3], Abdulwahab Almutairi[4] and Mohamed Abouhawwash[5,6]

[1] Department of Computer Science and Engineering, Anna University, University College of Engineering Dindigul, Tamilnadu, India

[2] Department of Computer Science and Engineering, CHRIST (Deemed to be University), Bangalore, India

[3] Department of Computer Science, CHRIST (Deemed to be University), Bangalore, Bangalore, Karnataka, India

[4] School of Mathematics, Unaizah College of Sciences and Arts, Qassim University, Saudi Arabia, Saudi Arabia

[5] Department of Mathematics, Faculty of Science, Mansoura University, Mansoura, Egypt

[6] Department of Computational Mathematics, Science, and Engineering (CMSE), Michigan State University, East Lansing, MI USA

Corresponding author
Abdulwahab Almutairi, almutairi2017@qu.edu.sa

## ABSTRACT

Cloud computing is one of the evolving fields of technology, which allows storage, access of data, programs, and their execution over the internet with offering a variety of information related services. With cloud information services, it is essential for information to be saved securely and to be distributed safely across numerous users. Cloud information storage has suffered from issues related to information integrity, data security, and information access by unauthenticated users. The distribution and storage of data among several users are highly scalable and cost-efficient but results in data redundancy and security issues. In this article, a biometric authentication scheme is proposed for the requested users to give access permission in a cloud-distributed environment and, at the same time, alleviate data redundancy. To achieve this, a cryptographic technique is used by service providers to generate the bio-key for authentication, which will be accessible only to authenticated users. A Gabor filter with distributed security and encryption using XOR operations is used to generate the proposed bio-key (biometric generated key) and avoid data deduplication in the cloud, ensuring avoidance of data redundancy and security. The proposed method is compared with existing algorithms, such as convergent encryption (CE), leakage resilient (LR), randomized convergent encryption (RCE), secure de-duplication scheme (SDS), to evaluate the de-duplication performance. Our comparative analysis shows that our proposed scheme results in smaller computation and communication costs than existing schemes.

Subjects Artificial Intelligence, Computer Networks and Communications, Computer Vision, Distributed and Parallel Computing, Security and Privacy
Keywords Cloud computing, Edge computing, IoT, Energy, Dynamic speed scaling algorithm, EIoT

## INTRODUCTION

Data redundancy is indirectly proportional to the data authentication. When redundancy increases, then the possibility of authentication of certain redundant data is much less. If data redundancy is reduced, possibility of data authentication is high. Deduplication has become a hot topic in recent years, in spite of the rapid growth in cloud computing and big data. The cost of cloud storage is highly reduced by using deduplication process which avoids storing of same data at multiple times in real time. Secured deduplication is provided by encrypting client data in the server. It must bring confidence among clients to believe service provider. Usually traditional techniques does not support deduplication with security. In this article, we implemented biometric techniques to ensure deduplication with data security.

### Cloud authentication issues

The advancement in data sharing and processing over the cloud makes consequence of innovative technologies like smart mobile devices, mobile applications with sensors, Internet spread, and usage, data availability in social media. These technologies make wider influences on big data in our day-to-day activity. Many organizations like Amazon, Flipkart and Netflix perform data collection, mining, and analysis from various sources. Sharing large volumes of data over the network has been made easily accessible through cloud storage. The increasing need for storage disks over the network searches for authentication of stored data has resulted in security concerns in the cloud and distributed storage. Large amounts of storage in cloud infrastructure are occupied by duplicate data records. Aiming to address these technical concerns, researchers have focused on techniques for data de-duplication using biometric de-duplication with user authentication. The link between intrinsic individual characters with their behavior, physical, physiological is used to authenticate an individual with biometric data recognition. While comparing with knowledge-based authentication, biometric can provide stronger security guarantees. Building a biometric-enabled technology on the cloud is important for safety as well as security enhancement. The security is provided to areas such as forensics, surveillance, defense, banking, and personal authentication. Further biometric-based authentication process has been proven to provide stronger security guarantees and robustness in contrast to traditional methods for sensitive applications (*Ah Kioon, Wang & Deb Das, 2013*)

### Cloud data redundancy issues

Data de-duplication is the process of avoiding redundant data copies reducing the overhead by eliminating duplicate stored data. Today's world is hyper-connected with online communication, payments, ticket services, using managed or unmanaged networks, and devices scaling across several endpoints. These devices are currently being protected with security technologies and enabled with cryptographic single factor, two factors, multi-factor authentication methods (*Alomar, Alsaleh & Alarifi, 2017*). Human in every communication uses multi-factor methods for fast, reliable, and user-friendly authentication while accessing online services. The digital age of information allows the

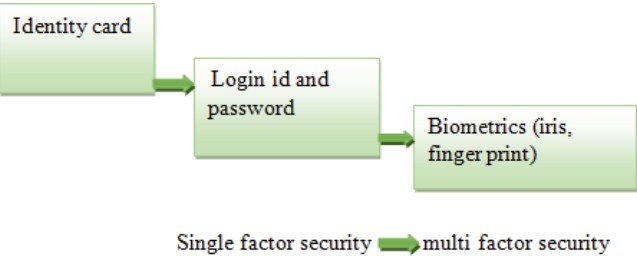

**Figure 1** **Security development stages.**

distribution and replication of data across the network. During this process, the same data may be shared, circulated, stored multiple times. This highlights the need for smart technologies to tackle the challenge of data deduplication along with authentication methods for users to access the data.

## Cloud data authentication for deduplication

This smart connected world makes the data secure with users through authentication processes (*Alomar, Alsaleh & Alarifi, 2017*). A user needs to identify himself in the system by sending authentication messages. When message 'A' is sent by the user, the system computes F(A) randomly and checks with stored data 'B'. A single authentication password alone cannot ensure the authentication of the user (*Benarous, Kadri & Bouridane, 2017*; *Mohsin et al., 2017*). Accessing sensitive data offline or online requires a fundamental security system for authentication (*Mohsin et al., 2017*; *Balloon, 2001*; *Ometov et al., 2018*) (Fig. 1). Traditional authenticated transactions like applying seals, wax seals are physical security systems (*Konoth, van der Veen & Bos, 2016*). Sender-based information validation alone cannot provide standard authentication (*Ibrokhimov et al., 2019*). Figure 1 shows how the technology is evaluated from single security techniques to multiple security techniques.

Initially, a single data factor was used for authentication, which was eventually compromised by the research community (*Kim & Hong, 2011*; *Dasgupta, Roy & Nag, 2016*). Examples of single-factor security are user ids and passwords. Password authentication is considered the weakest level of security (*Bonneau et al., 2015*; *Wang & Wang, 2015*). Sharing of security information like a password can easily lead to compromised security. Several attacks have occurred by unauthorized entities like dictionary hacking (*Ah Kioon, Wang & Deb Das, 2013*), rainbow attacks (*Heartfield & Loukas, 2015*), and other social engineering attacks (*Grassi et al., 2016*). When users choose password-based authentication, the complexity of the authentication must be ensured (*Gunson et al., 2011*). High protection of the accounts cannot be ensured by using a single authentication factor (*Sun et al., 2014*). The next level involves a two-factor authentication, which is achieved by an identity or security question (*Bruun, Jensen & Kristensen, 2014*; *Harini & Padmanabhan, 2013*). At present, three categories of groups are available for connecting individuals with security credentials (*Scheidt & Domangue, 2006*):

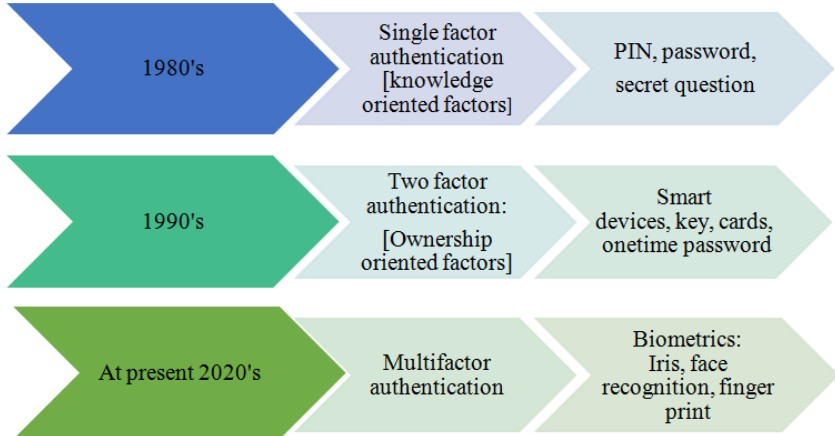

**Figure 2 Evolution of various authentication levels.**

- Ownership authentication –requires ID cards, smartphones.
- Knowledge authentication –requires passwords, secret keys.
- Biometric authentication –biometric data (fingerprints, iris scans, face scans)

Multi-factor authentication provides and ensures higher safety levels, with two or more credentials (*Bhargav-Spantzel et al., 2007*; *Banyal, Jain & Jain, 2014*; *Council & Committee, 2010*). Biometrics is widely used in the multi-factor authentication process based on individual biological and behavioral characteristics (*Huang et al., 2014*). The higher level of security is offered explicitly by biometrics recognition using more security factors (*Tahir & Tahir, 2008*) and the evolutionary history of the authentication is described in Fig. 2.

High-security infrastructures utilize multiple authentication factors for protecting the information. For example, ATM cash withdrawal has a combination of ownership patterns (card) which is accessed by using knowledge factors, such as PIN, to transfer money and manage accounts (*Coventry, De Angeli & Johnson, 2003*; *Ometov et al., 2018*). To make this system even more robust, a card with a PIN is further authenticated using a one-time password, while accessing the sensitive data (*Aloul, Zahidi & El-Hajj, 2009*). Facial recognition methods are also suggested for user authentication purposes (*Ometov et al., 2018*). In a recent survey, it has also been pointed out that most business enterprises choose multi-factor-based security and authentication for the secure processing of transactions. At present, most enterprises use biometric systems for employee verification, bank management, lockers, vehicles, etc., making the multi authentication factor stronger and interesting, which paved the way to introduce innovative biometrics, among the research community.

Most of the electronic devices use multiple authentication techniques for security-based access and allow only authenticated owners to use the devices (*Symeonidis, Mustafa & Preneel, 2016*). One such potential application could be the usage of biometrics in a

vehicle to authenticate its owner. With respect to market-based applications, authentication techniques can be broadly categorized as commercial, governmental, and forensic applications. Commercial purposes may include account access and ATM banking. Government needs may include document identification, government ID process, social security applications, military, border security controls. Lastly, forensic applications may include investigations, evidence identification, criminal identification (*Liu et al., 2018*; *Nor et al., 2015*; *Grigoras, 2009*).

One of the challenges in multi-factor authentication techniques is the association between users and the sensors being used (*Sasi & Saranyapriyadharshini, 2015*). Regarding security, this relationship must be established so that only authorized users, authenticated in advance, should receive access rights. In this context, the use of biometrics (e.g., fingerprints, face recognition) is considered a user-friendly technique.

### Contribution

In this research work, a multi-factor authentication technique with biometrics is proposed for the verification of users in cloud environments. The bio key of the data owner is first generated for the data stored in the cloud. Contributions on security and data redundancy are addressed in this paper through the following components:

1. A multi-factor authentication technique is used for bio key generation. Finger print of owner is processed for selecting appropriate features using edge detection with hashing function. Bio key generated based on extracted feature set.
2. We newly introduce bio key from the data owner's fingerprint. It is chosen for the generation of bio keys which helps data to be shared with owners knowledge.
3. The encrypted data is stored in the cloud so that the data can be accessed by the user only when the bio key shared for the authentication process is satisfied.
4. The redundancy of stored data is eliminated while keeping data security in mind.

This article is organized into five sections. 'Literature Review' presents a literature survey and 'Proposed Methodology: Bio Key With GABOR –XOR' presents the proposed methodology and its design. 'Experimental Results and Discussions' describes the evaluation results of the proposed method and finally, 'Conclusion' concludes our paper and discusses future work.

## LITERATURE REVIEW

*Zareen et al. (2017)* proposed a secure biometric authentication scheme for user identification and mutual authentication using elliptic curve cryptography for key generation and key exchange with optimal communication cost. *Kathrine et al. (2017)* proposed a cloud-based mobile biometric authentication framework (BAM Cloud) using dynamic signatures and user authentication. The data is captured with a handheld mobile device and subsequently, storage, preprocessing, and training are done in a distributed manner on the cloud. The proposed method was implemented using Map Reduce on the Hadoop platform and for training a Levenberg–Marquardt backpropagation neural network model was used, achieving a speedup of 8.5× and an accuracy of 96.23

*Al-Assam, Hassan & Zeadally (2019)* surveyed various biometric-based authentication methods in cloud environments. The traditional password-based authentication lacks security when it comes to cloud data. To this end, multifactor authentication is suggested which allows two or more authentication parameters along with password-based security. This review focuses on the various available biometric authentication models and their advantages and disadvantages.

*Wong & Kim (2012)* provided the concepts used in biometric-based authentication in cloud computing. They discussed the challenges and limitations of traditional methods, along with attack scenarios, such as misuse of biometric data, to track individuals and leak confidential information related to health, gender, ethnicity, etc. At the same, they also argued that the privacy of cloud-based biometric authentication is not going to resolve the authentication issues technically, rather new legislation to enforce privacy-aware measures on cloud service providers related to the biometric collection, data processing, and template storage is opined, leading secure authentication in cloud environments.

*Dulari & Bhushan (2019)* proposed a authentication method for user in cloud computing based distributed environment. In this process users biometric information are stored as template in cloud server. Further user verification is done with several participants. Here users feature vector query is compared with template saved in cloud server. In this method, homomorphic based encryption is used for matching the protocol. This matching protocol helps to compare the queried vector with template available as encrypted file . The metrics used for measuring output are Square of Euclidean distance, sensitive preservation of information and processing of authentication with high security .

*Dulari & Bhushan (2019)* proposed a security system called TORDES for cloud storage. This work uses legion containers in cloud storage. The data is stacked in TORDES for authentication with crypto-biometric systems to avoid unauthorized access.(*Indu, Anand & Bhaskar, 2018*). surveyed the different biometrics applied on cloud security issues to identify malware. *Ziyad & Kannammal (2014)* analyzed cloud security and threat possibilities. They analyzed several security mechanisms and suggested the robust ones for both academic and industry environments. *Ziyad & Kannammal (2014)* proposed a multifactor biometric authentication system for cloud computing. The biometric features considered in this work were palm vein and fingerprints, where palm vein biometric data was stored in multicomponent smart cards and fingerprint data in the central database of a cloud server.

*Zahrouni et al. (2017)* developed an application that serves as an extra layer of security on top of a pre-existing banking application by using a Biometric lock application, which allows a user to add a layer of security to their credit and debit cards, at the expense of minimal time overhead. *Malathi & Raj. R (2016)* proposed a biometrics-based user identification scheme using the features of a palm print, fingerprint, and iris for providing accurate personal identifications.

*Wang et al. (2018)* carried over *Amin et al. (2018)* protocol as a case study to provide ideas for designing secure protocols for cloud environments in order to overcome existing security weaknesses in the protocol. They further improved the protocol using BAN logic and also used heuristic analysis to prove the security of the protocol. *Obergrusberger et al.*

**Table 1 Literature survey of existing techniques.**

| Authors | Problem | Methodology | Advantages | Integrity | Confidentiality |
|---|---|---|---|---|---|
| *Shin et al. (2020)* | Data deduplication and security in MEC | server less efficient encrypted deduplication (SEED) +Lazy encryption | It takes 1000ms for processing128MB | No | yes |
| *Wang, Wang & Zhang (2019)* | Double payment avoidance and data deduplication | Block chain technology | Avoids third party | yes | yes |
| *Wu et al. (2019)* | File deduplication and integrity | Confidentiality preserving deduplication using public auditing (CPDA) | Computation of authentication tag is done by CSP | No | Yes |
| *Shen, Su & Hao (2020)* | Deduplication and brute force dictionary attacks | Light weight cloud storage auditing for deduplication | light-weight computation on the user side | yes | No |
| *Liu et al. (2020)* | Data deduplication in files | Verifiable ABKS over encrypted cloud data is proposed | realize indistinguishable keywords, unforgetability of signature and confidentiality of cipher texts | No | Yes |
| *Bai, Yu & Gao (2020)* | cloud storage auditing and deduplication literatures fail to support the modifications of ownership | re-encryption algorithm and the secure identity-based broadcast encryption technology | ownership modification and integrity is maintained | Yes | No |
| *Shynu et al. (2020)* | Data deduplication | Modified Elliptic Curve Cryptography (MECC) algorithms | Recognize data redundancy at the block level | No | Yes |

*(2012)* proposed biometric observer techniques and provides a idea on fundamental trust rules in web as a prototype for implementation . The enrollment of users are supervised by an observer. further observers are those who enhance to provide authentication to biometric template. It provides more trustworthiness model for biometric identities. strong trust is build between observers and other due to best relation between both observers and individuals observed in the Database of system. *Chandramohan, Vengattaraman & Dhavachelvan (2017)* proposed a privacy-preserving model to prevent digital data loss in the cloud, helping the cloud requester/users) to trust their proprietary information and data stored in the cloud. Table 1 below represents the recent deduplication research problems and their methodology to overcome the identified problem. It is noticed that biometric based deduplication and authentication of data is not much used recently. Our proposed work shows advanced deduplication handling process with high security.

*Shabbir et al. (2021)* noticed profits of Mobile Cloud Computing (MCC) in medical healthcare. It faces more challenges in security and privacy of customer data. Here they implement layered security modeling using Modular Encryption Standard(MES) to increase the security of MCC. The performance is better than other encryption techniques. *Rehman et al. (2021)* addressed invehicle communication problems using controller area network and electronic control units. Security during communicating inside the vehicle is tackled using novel approach CANintelliIDS. It detects the vehicle

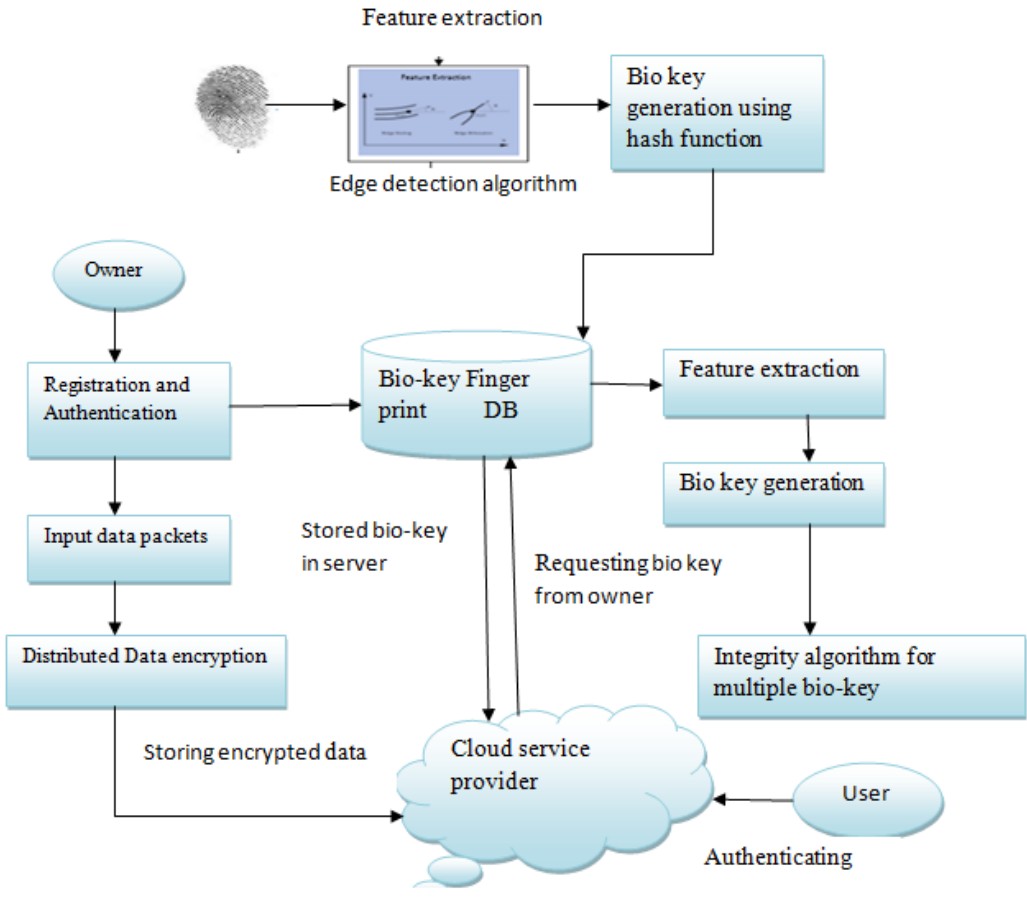

**Figure 3  Overview of the proposed architecture.**

intrusion attack across controller area network. at result it gains 10.79 *Naeem et al. (2021)* discusses energy efficient WSN for high performance network lifetime. In this article hybrid technique called Distance aware residual energy-efficient stable election protocol is used with energy efficient election protocol for optimal transmission routes. In this outcome energy efficiency is increased for 10.

## PROPOSED METHODOLOGY: BIO KEY WITH GABOR –XOR

In distributed cloud computing, biometric-based authentication plays a vital role in current research. Distributed denial of service is the main threat in the cloud nowadays: several users trying to access a single cloud server leads to an increase in response time and complicates security. There are several methods to solve these issues in the cloud, even though there is a lack of confidentiality, reliability, and consistency of the data. To resolve that, we propose an approach called secure biometric authentication on distributed storage in the cloud. Figure 3 shows the overview of the proposed architecture. The owner's biometric information is recorded for authentication. Once the owner registration is complete, the data is encrypted using a distributed approach and is stored

in the cloud. While the user tries to access the content, a cloud server authenticates the user validity and contacts the data owner for the bio key to access the data. In the proposed architecture, the data is stored securely and with the owner's reference, users can access the content avoiding duplicate copies of the same content and enhancing security through the bio key. The flow of the proposed design can be described as:

- The owner of the data can upload content onto the cloud that is encrypted through a distributed model.
- The duplication of the content can be avoided by the cloud server. The features of the fingerprint of the owner are extracted and stored in a database so that the original content can be referred by the user with the owner's permission.
- Finger print of the user is converted in to bio key using edge detection and hash functions.

## Biometric based authentication with de-duplication

In this work, a fingerprint-based authentication along with a hash-based deduplication approach is used. Among the various biometric techniques, fingerprints have been widely accepted and implemented for secure authentication. The input fingerprint image of the owner is normalized before processing and then the features of the fingerprint image are extracted using an optimized self-learning method and are stored in a database for authentication. Figure 3 shows the biometric authentication process. A Gabor filter-based technique is used to enhance image equality by removing the noise. Subsequently, the enhanced image is made ready for feature extraction (*Hur et al., 2016*). The ridge endings and ridge bifurcation are extracted as features using an edge detection algorithm then the hash functions of these features are considered as bio-keys, which are stored in a database for authentication and de-duplication.

## Security based –distributed storage encryption

This proposed algorithm is taken from *Kumar & Begum (2011)*, where the input data packet is divided into two substrings. The substrings are further processed and then merged to store in the cloud server. The two components of the data packets are considered as X , Y and Z is the random number. To encrypt the data packet, an data packet an XOR operation is performed and the data is encrypted that is sent to the cloud server. This approach is shown in Fig. 4. Hence, our proposed biometric security-based-distributed storage encryption with a de-duplication approach can obtain secure data transfer between the user and the cloud without duplication. This proposed approach can avoid duplication of content by sharing the key to the user for access. This will leads to reduce the storage of the cloud. One method can prove the secured authentication using biometric and de-duplication.

**Algorithm 1:** Biometric authentication with de-duplication

**Result**: Bio key (QR-code)

**Input:** Input fingerprint image;

---

**Step 1:** read the input image with the pixels as I(x,y);

**Step 2:** The input image can be preprocessed (e.g., binarization, thinning) and enhanced using the following equations. During binarization, the grey level image is converted into a black and white image, where black represents the ridges and white represents the valleys. Then the Binarized image is thinned using three conditions. Thinning is the process of transforming the ridge pixels into one pixel:;

$$I(x,y) = \frac{1}{|w||h|} \sum_{i=0} \sum_{j=0} (x,y) \qquad (1)$$

$$g(x,y;\theta,f) = exp\left\{-\frac{1}{2}\left[\frac{x\theta}{\sigma^2}\right]cos(2\pi fx^2))\right\} \qquad (2)$$

Where, $x_\theta = xcos_\theta + ysin_\theta$ and $y_\theta = -xsin_\theta + ycos_\theta$, $\theta = Orientation$, f=frequency and $\sigma_x, \sigma_y$=standard deviation of gausian envelop;

**Step 3:** The features like ridge end and bifurcation are extracted using the following equation;

$$\sum_{i=0}^{7} N_i = thenridgeend \qquad (3)$$

$$\sum_{i=0}^{7} N_i > 2thenbifurcation \qquad (4)$$

Where $N_0, N_1..N_7$ are the eight neighbors of the pixel (x, y) of Image I. ;

**Step 4:** the binary pattern code of the processed image is generated as follows;

$$BC(g(x_c,y_c)) = \sum_{i=0}^{7} f(gx_i) - g(x_c)x2^n \qquad (5)$$

Where, f (m) = 1 for $m >= 0$ and f (m) =0 for $m < 0$;

**Step 5:** The bio key for each binary pattern is generated with QR decomposition formula [2];

**Step 6:** De-duplication using a hash-based technique of the key as;

$$KH_i = hash(KH_i|G_k), KH_j \qquad (6)$$

Where $G_k$- Gaussian random number.;

**Step 7:** Store the bio key of each owner into the cloud data base and content storage using algorithm 2.

---

**Algorithm 2:** P

**Result**: Encrypted data packet

**Input:** Data packet with name label NL and pre-defined label PL;

---

**Step 1:** The name label of the data packet is { $n_1$ ,$n_2$, , $n_l$ } and pre-defined label are declared as { $p_1$, $p_2$,, $p_l$ };

**Step 2**;

**while** *NL* **do**

    **while** *each packet of the input data* **do**

        **if** *lable <> PL* **then**

            Initialized and J,$\gamma$, $\delta = 0$;

            Generate key $k$ at random;

            **if** *(I&&Z!=0)* **then**

                J=Z-I;

                $\gamma$=I XOR k;

                $\delta$=J XOR k

            **end**

        **else**

            Encrypt the data packet using XOR operation;

        **end**

    **end**

**end**

**Step 3:** Output the encrypted data packet. That will store into the cloud server;

**Step 4:** De-duplication: algorithm 1 and algorithm 2 are combined.;

**Note:** The step 4 combines two algorithms and computes when user request for data, cloud server authenticate it and search for owner of the data. Finally it shared the bio key to the user to access the content. ;

seudo code : 2 (Distribute storage security based encryption)

---

## EXPERIMENTAL RESULTS AND DISCUSSIONS

This proposed work experimented in AWS cloud services. Different type of services of AWS is initiated to each owner and user for a secure authentication process. The proposed work is implemented in three stages. The first stage is to generate the bio key of the owner of the data before upload. In the second stage, the hash function of this key is used for the de-duplication process. The third stage is the encryption of the data that are uploaded to the cloud server. Figure 5 represents the first stage of the proposed work. In Fig. 5A, represents the original input fingerprint image of the owner and (Fig. 5B) and (Fig. 5C) represents the equivalent orientation and frequency images of the input image. This is shown in Fig. 6.

**Illustration of Fig. 6:** It illustrates the Binarized form of the input image to identify the ridges and valleys. Then the Binarized image is thinned into a single pixel to identify the

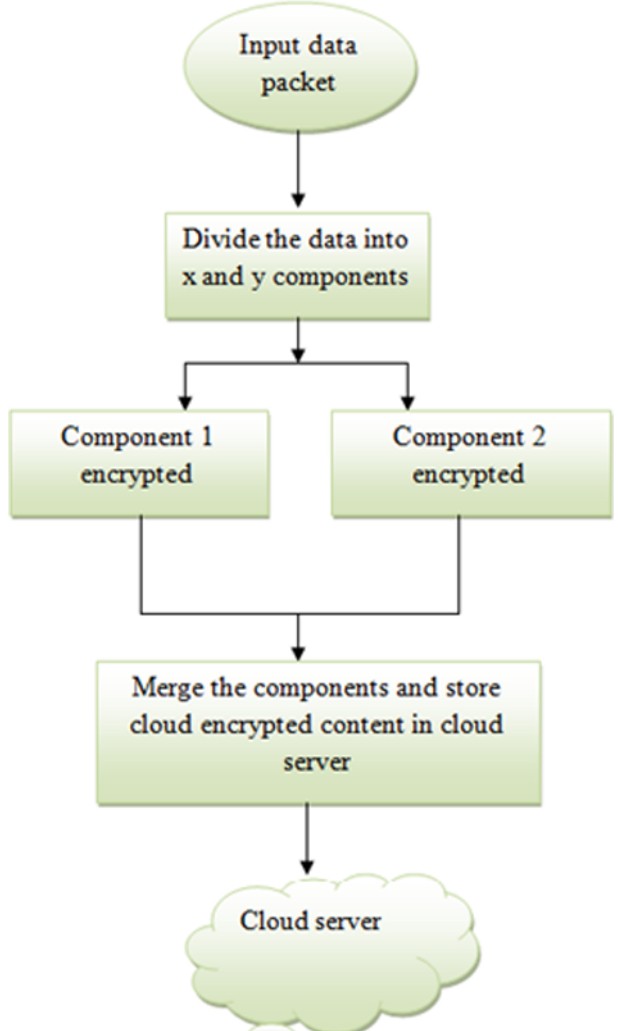

**Figure 4   Security based distributed storage encryption.**

features such as ridge ending and bifurcation which is shown in Fig. 6C. These variations of the fingerprint patterns are used to build the bio key with QR code as shown in Fig. 6D.

The comparative analysis of time is shown in Fig. 7. The data redundancy makes memory engaged and takes more time to upload, download the data. While we upload, redundancy makes no space for new data and resource allocation takes more time. Next, while we download the data, due to redundancy system confuses which data to access. Memory space also very less. This makes more time to download the data from a cloud server.Also, Fig. 8 shows time taken by proposed model in data encryption and decryption. To evaluate the performance of this fingerprint authentication de-duplication, the proposed work is compared with existing algorithms such as Local Binary Pattern (LBP), Scale Invariant Feature Transform (SIFT), Manifold Learning, and cloud user to service authentication (CU-SA) on different data size. The evaluated results are

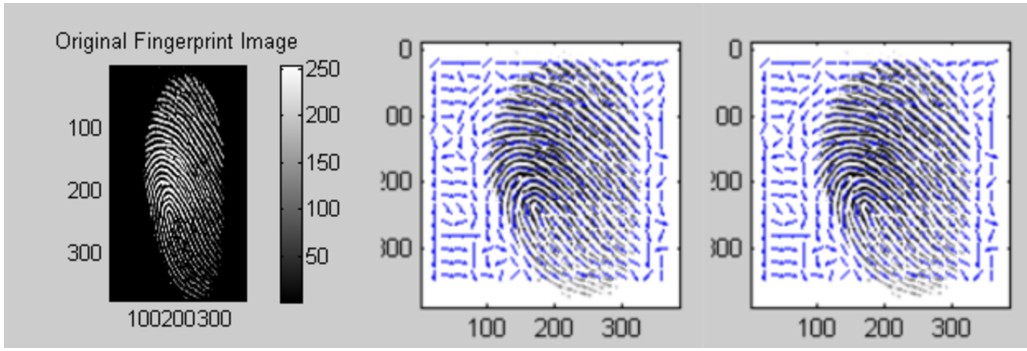

**Figure 5** (A) Original fingerprint image (B) ridge orientation (C) ridge frequency.

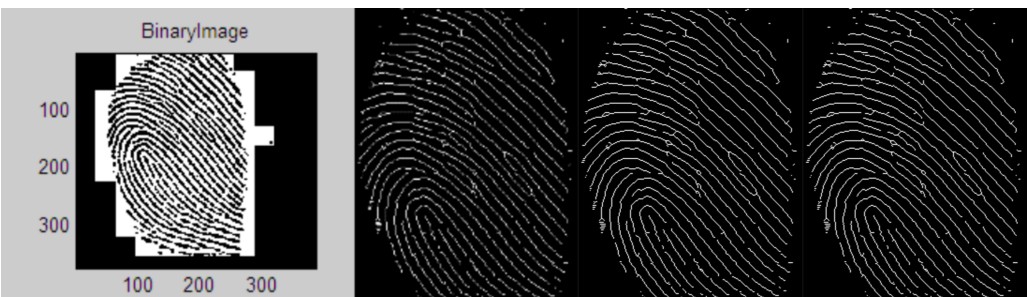

**Figure 6** (A) Binarization (B) thinned image (C) feature (Ridge & bifurcation) extraction (D) QR code of the fingerprint input image.

shown in Fig. 8. Proposed approach consumes only 890 ms for 100 DB size. Whereas other approach exceeds nearly 1000 ms and above. From the resultant observation, the proposed biometric authentication de-duplication is best in terms of computational time by having low time compare to other standard existing algorithms.

To evaluate the proposed biometric security-based distributed storage authentication de-duplication approach, the encryption and decryption using the bio key are calculate with communication cost for different data sizes. The evaluated results are shown in Figs. 9 and 10. In the encryption process, our proposed algorithm takes 5 ms, 8 ms, 13 ms, 23 ms, and 26 ms for encrypting data with the size of 10 MB, 30 MB, 50 MB, 80 MB, and 100 MB. Similarly, for the decryption process, our proposed algorithm takes 4 ms, 7 ms, 12 ms, 20 ms, and 33 ms for encrypting data with the size of 10 MB, 30 MB, 50 MB, 80 MB, and 100 MB.

From all observations of the result experiments, the proposed bio key de-duplicating with security-based distributed storage approach performs better encryption and decryption time and communication cost with respect to different data size. The proposed approach is compared with the existing algorithms such as convergent encryption (CE) (*Liu et al., 2017*), leakage resilient (LR) (*Liu et al., 2017*), randomized convergent encryption (RCE) (*Liu et al., 2017*), secure de-duplication scheme (SDS) (*Liu et al., 2017*)

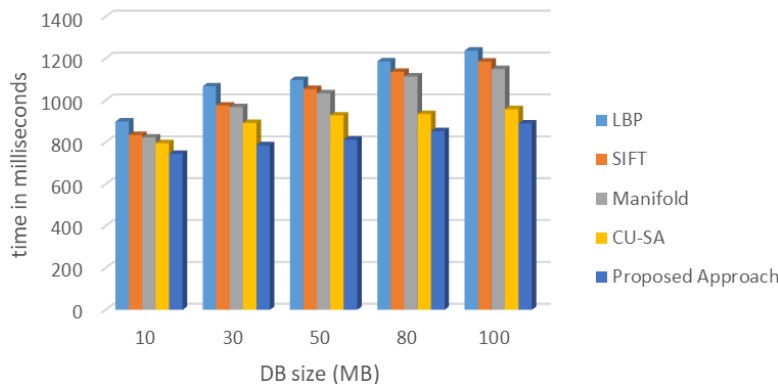

**Figure 7 Evaluation of computational time of various algorithms.**

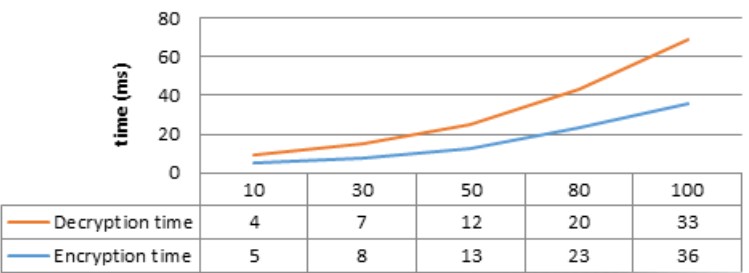

**Figure 8 Proposed method evaluation in terms of encryption and decryption time.**

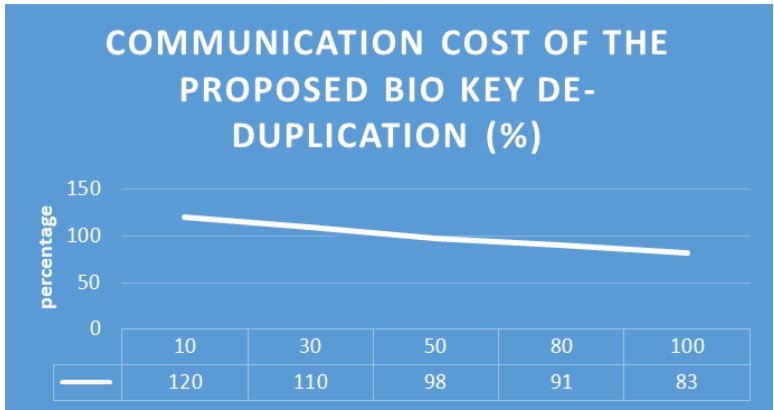

**Figure 9 Proposed method communication cost.**

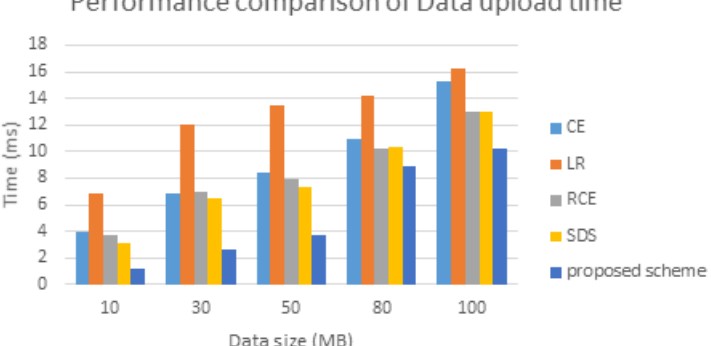

**Figure 10  Performance comparison in terms of data uploads time.**

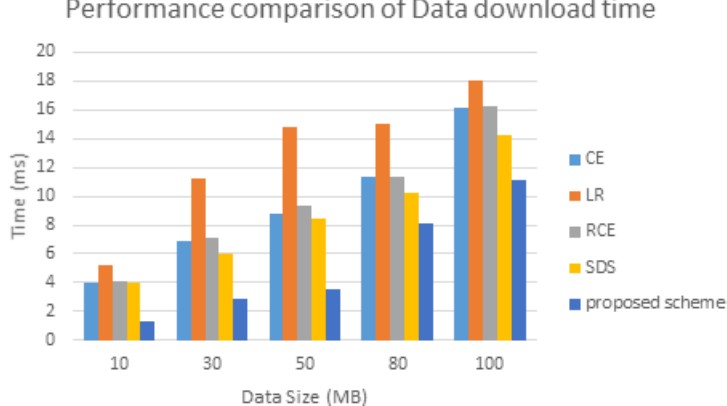

**Figure 11  Performance comparison of data downloads time.**

to prove the de-duplication performance. The Fig. 9 describes the communication cost of proposed system. The evaluated experiment in terms of the data upload to the cloud server and the data download from the server of various algorithms are shown in Figs. 10 and 11. The obtained result proved that our proposed bio key-based encryption and de-duplication techniques secure minimum upload and download time on various sizes of the data compared to other algorithms. Hence our proposed scheme achieves better communication cost, better encryption and decryption time, and better data upload and downloads time. Which proved that one method can be used for secure authentication, de-duplication with excellent reliability, response time, and security.

## CONCLUSION

Cloud storage has abundance data at every second for processing and storing in server by CSP. Data deduplication in the cloud brings concern on security of stored data. There is lot of possibilities for unauthorized access. In our proposed work, security using bio key generation makes users access data securely. Here, the main task of accomplishment is

avoiding redundancy in a cloud server and biometric authentication using Gabor filters with XOR operation. This is a very complex fact due to biometric scanning and matching for authentication. The evolution of high technologies over time assures the security and integrity of the system. Also, data de-duplication is reducing multiple storages of the same data over the cloud network. In this article, we have initiated security with the de-duplication of the data in the cloud servers. Security of the data is computed using the user's biometric parameters and a bio key is generated. The fingerprint of the owner is used to generate the bio key by QR code conversion. User is authenticated by transferring the bio key to users. Then the data is transferred to authenticated users. The most significant task carried out in this research work is biometric cryptographic security and reducing the de-duplication of data in cloud storage. The algorithm and its processing time are improved in our implementation process. The proposed methodology provides more reliability and fast encryption techniques. In the future, different biometric techniques can be expanded for user authentication processes. Standardized intelligent algorithms can be further used for checking data deduplication.

### Funding
The authors received no funding for this work.

### Competing Interests
The authors declare there are no competing interests.

### Author Contributions
- Vinoth Kumar M conceived and designed the experiments, prepared figures and/or tables, and approved the final draft.
- K Venkatachalam conceived and designed the experiments, analyzed the data, prepared figures and/or tables, and approved the final draft.
- Prabu P performed the experiments, analyzed the data, authored or reviewed drafts of the paper, and approved the final draft.
- Abdulwahab Almutairi and Mohamed Abouhawwash performed the experiments, performed the computation work, authored or reviewed drafts of the paper, and approved the final draft.

### Data Availability
    The data and code are available at GitHub:
    - https://github.com/prabukiwi1/Biometric-Code.
    - https://github.com/prabukiwi1/Biometric-Datasets.

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
