# Peer review of "Secure biometric authentication with de-duplication on distributed cloud storage"

_PeerJ Computer Science, doi:10.7717/peerj-cs.569_

## Round 0.1 · original submission · Major Revisions

Dear Dr. Venkatachalam,
Thank you for your submission to PeerJ Computer Science.

It is my opinion as the Academic Editor for your article - Secure biometric authentication with de-duplication on distributed cloud storage - that it requires a number of Major Revisions.

My suggested changes and reviewer comments are shown below and on your article 'Overview' screen. Please address these changes and resubmit.

·

Basic reporting

Grammar and typo mistakes are there.

Experimental design

no comment

Validity of the findings

no comment

Additional comments

Introduction should be revised: (1) Introduce the problem (2)discuss about some of the existing solutions (3)identify the gap or scope of improvement (4) discuss in order to address the identified gaps what is the methodology used (5) list out the contributions.
Literature review should have a table to summarize research gap in each of the existing solutions.
Literature review/Introduction should include recent references such as:
An enhanced approach for three factor remote user authentication in multi - server environment
A Review of Machine Learning Algorithms for Cloud Computing Security
Enabling Secure Authentication in Industrial IoT with Transfer Learning empowered Blockchain

Most figures are of poor quality. What's the use of Fig. 1?
Can Fig. 3 and Fig. 4 be integrated to depict the proposed work? Also, explain the same step-by-step.
All graphs should be uniform and analyzed in detail.
Conclusion should not reflect Abstract and should conclude the work.

Reviewer 2 ·

Basic reporting

A biometric authentication scheme is proposed for the requested users to give access permission in a cloud-distributed environment and, at the same time, alleviate data redundancy. To achieve this, a cryptographic technique is used by service providers to generate the bio-key for authentication, which will be accessible only to authenticated users. A Gabor filter with distributed security and encryption using XOR operations is used to generate the proposed bio-key (biometric generated key) and avoid data deduplication in the cloud, ensuring avoidance of data redundancy and security.

Experimental design

Satisfactory.

Validity of the findings

Satisfactory.

Additional comments

1. Check the paper thoroughly for grammatical mistakes. There are some grammatical errors which have to be rectified.
2. What are the main contributions of the current work?
3. The literature survey section can be summarized as a table.
4. Some of the recent and relevant works such as the following can be discussed in the paper: "Load balancing of energy cloud using wind driven and firefly algorithms in internet of everything, A systematic review on clone node detection in static wireless sensor networks, IMCFN: Image-based malware classification using fine-tuned convolutional neural network architecture".
5. A detailed analysis on the results achieved has to be discussed in the results section.
6. Discuss the limitations of the current work in conclusion.

Reviewer 3 ·

Basic reporting

- Please highlight the contribution clearly in the introduction
-The author should highlight the contribution clearly in the introduction and provide a comparison note with existing studies.
- Some Paragraphs in the paper can be merged and some long paragraphs can be split into two.
- The quality of the figures can be improved more. Figures should be eye-catching. It will enhance the interest of the reader. e.g., "Figure 1. Security development stages", "Figure 3. Overview of the proposed architecture"
- Authors should add the most recent reference:
1) https://ieeexplore.ieee.org/abstract/document/9316223
2) https://ieeexplore.ieee.org/abstract/document/9359538
3) A. Naeem, A. R. Javed, M. Rizwan, S. Abbas, J. C. -W. Lin and T. R. Gadekallu, "DARE-SEP: A Hybrid Approach of Distance Aware Residual Energy-Efficient SEP for WSN," in IEEE Transactions on Green Communications and Networking, doi: 10.1109/TGCN.2021.3067885.

- please give a proofread check to the paper.

Experimental design

No comments

Validity of the findings

- What are the computational resources reported in the state of the art for the same purpose?
- What are the evaluations used for the verification of results?
- Clearly highlight the terms used in the algorithm and explain them in the text.

Additional comments

- Please cite each equation and clearly explain its terms.

---

## Round 0.2 · accepted · Accept

Dear Dr. Venkatachalam,

Thank you for your submission to PeerJ Computer Science.

I am writing to inform you that your manuscript - Secure biometric authentication with de-duplication on distributed cloud storage - has been Accepted for publication. Congratulations!

Reviewer 2 ·

Basic reporting

In this article, a biometric authentication scheme is proposed for the requested users to give access permission in a cloud-distributed environment and, at the same time, alleviate data redundancy. To achieve this, a cryptographic technique is used by service providers to generate the bio-key for authentication, which will be accessible only to authenticated users. A Gabor filter with distributed security and encryption using XOR operations is used to generate the proposed bio-key (biometric generated key) and avoid data deduplication in the cloud, ensuring avoidance of data redundancy and security.

Experimental design

Satisfactory.

Validity of the findings

Satisfactory.

Additional comments

The authors have addressed all the comments. I accept the paper in its present form.

Reviewer 3 ·

Basic reporting

The authors have addressed my suggestions. I would like to accept this paper.

Experimental design

The authors have addressed my suggestions. I would like to accept this paper.

Validity of the findings

The authors have addressed my suggestions. I would like to accept this paper.

Additional comments

The authors have addressed my suggestions. I would like to accept this paper.